# Antigen-specific modulation of chronic experimental autoimmune encephalomyelitis in humanized mice by TCR-like antibody targeting autoreactive T-cell epitope

Alona Goor, Efrat Altman, Inbar Arman, Shir Erez, Maya Haus-Cohen, Yoram Reiter 

The development and application of human TCR-like (TCRL) antibodies recognizing disease-specific MHC–peptide complexes may prove as an important tool for basic research and therapeutic applications. Multiple sclerosis is characterized by aberrant CD4 T-cell response to self-antigens presented by MHC class II molecules. This led us to select a panel of TCRL Abs targeting the immunodominant autoantigenic epitope MOG$_{35-55}$ derived from myelin oligodendrocyte glycoprotein (MOG) presented on HLA-DR2, which is associated with multiple sclerosis (MS). We demonstrate that these TCRL Abs bind with high specificity to human HLA-DR2/MOG$_{35-55}$-derived MHC class II molecules and can detect APCs that naturally present the MS-associated autoantigen in the humanized EAE transgenic mouse model. The TCRL Abs can block ex vivo and in vivo CD4 T-cell proliferation in response to MOG$_{35-55}$ stimulation in an antigen-specific manner. Most significantly, administration of TCRL Abs to MOG$_{35-55}$-induced EAE model in HLA-DR2 transgenic mice both prevents and regresses established EAE. TCRL function was associated with a reduction in autoreactive pathogenic T-cell infiltration into the CNS, along with modulation of activated CD11b+ macrophages/microglial APCs. Collectively, these findings demonstrate the combined action of TCRL Abs in blocking TCR-MHC interactions and modulating APC presentation and activation, leading to a profound antigen-specific inhibitory effect on the neuroinflammatory process, resulting in regression of EAE. Our study constitutes an in vivo proof of concept for the utility of TCR-like antibodies as antigen-specific immunomodulators for CD4-mediated autoimmune diseases such as MS, validating the importance of the TCR-MHC axis as a therapeutic target for various autoimmune and inflammatory diseases.

## Introduction

Multiple sclerosis (MS) is a chronic neuroinflammatory autoimmune disease of the central nervous system (CNS) characterized by immune cell infiltration across the blood–brain barrier (1, 2, 3, 4, 5). This infiltration promotes inflammation, demyelination, gliosis, and neuroaxonal degeneration, disrupting neuronal function. Early in lesion formation, autoreactive CD4 T lymphocytes play a crucial role by mounting aberrant responses against CNS autoantigens. These autoreactive CD4 T cells are activated by innate immune cells, including DCs, natural killer (NK) cells, macrophages in the periphery, and microglia, astrocytes, and infiltrating monocytes in the CNS (1, 2, 3, 4, 5). These cells mediate damage through antigen presentation on MHC class II molecules, presenting CNS autoantigens to autoreactive T cells. The strong genetic association of MS with the HLA-DRB1*15:01 allele underscores the importance of MHC class II in the disease's pathogenesis (6, 7).

Current immunomodulatory drugs for MS effectively reduce immune cell activity but have systemic effects and are often associated with significant side effects, such as flu-like symptoms and progressive multifocal leucoencephalopathy. Therefore, there is an unmet need for more specific therapies that can eliminate autoreactive immune responses without broadly compromising the immune system (8).

Decades of clinical and basic experimental research into multiple sclerosis point to distinct immunological pathways that drive disease relapses and progression. The variation in clinical manifestations of the disease correlates with the spatiotemporal dissemination of lesioned sites of pathology within the CNS. These lesions are a hallmark of multiple sclerosis and are caused by immune cell infiltration across the blood–brain barrier that promotes inflammation, demyelination, gliosis, and neuroaxonal degeneration, leading to disruption of neuronal functions (1, 2, 3, 4, 5). T cells appear early in lesion formation, and the disease is considered to be autoimmune, initiated by autoreactive lymphocytes that mount aberrant responses against CNS autoantigens. Intervention of the infiltration of immune cells from the periphery into the CNS has been the main target of currently available therapies for multiple sclerosis. Although broad-spectrum immunomodulatory drugs reduce immune cell activity and entry into the CNS and decrease relapse frequency, they are often associated with side effects. These

Laboratory of Molecular Immunology and Immunotherapy, Faculty of Biology, Technion-Israel Institute of Technology, Haifa, Israel

Correspondence: reiter@technion.ac.il

range from flu-like symptoms and the development of other autoimmune disorders to malignancies and even fatal opportunistic infections such as progressive multifocal leucoencephalopathy (8). Thus, more specific therapeutic targets that can be efficaciously modulated without inducing such significant adverse reactions are required.

The immune cells that characteristically infiltrate from the periphery into the CNS in MS must be specifically activated in order to induce the associated tissue damage. During the establishment of central tolerance in the thymus, most autoreactive T cells are deleted; however, this process is imperfect, and some autoreactive T cells are released into the periphery. In healthy population, peripheral tolerance mechanisms keep these cells in check. If this tolerance is decreased by reduced function of regulatory T (Treg) cells and/or the increased resistance of effector B cells and T cells to suppressive mechanisms, specific autoreactive B cells and T cells can be activated in the periphery to become aggressive effector cells. This activation may be in the form of molecular mimicry, novel autoantigen presentation, recognition of sequestered CNS antigens released into the periphery, or bystander activation. Genetic and environmental factors, including infectious agents and cigarette smoke constituents, contribute to these events. Once activated, differentiated CD4$^+$ T helper 1 (TH1) and TH17 cells, CD8$^+$ T cells, B cells, and innate immune cells can infiltrate the CNS, leading to inflammation and tissue damage. B cells trafficking out of the CNS can undergo affinity maturation in the lymph nodes before re-entering the target organ and promoting further damage. In the common experimental model of MS, experimental autoimmune encephalomyelitis (EAE)–infiltrating CD4$^+$ T cells are re-activated in the CNS by APCs, including CD11c+ DCs, with the resulting inflammatory response leading to monocyte recruitment into the CNS, and naïve CD4$^+$ T-cell activation through epitope spreading that further fuels the inflammation (2, 3, 4, 5). Autoantigen presentation by APCs to CD4$^+$ T cells is thus the epicentre of disease initiation and progression, and the MHC:TCR interaction between the APCs and the pathogenic autoreactive T cells is the central driving force of immune activation.

This autoantigen presentation of class II HLA-DR molecules to the pathogenic autoreactive TCR on T cells is thus a most specific checkpoint in the inflammatory process. We therefore aimed to block these specific MHC: TCR interactions.

The principle of our experimental design is to generate a T-cell receptor–like (TCRL) antibody that mimics TCR specificity binding to human MHC class II complexes presenting a myelin-derived autoantigen peptide on APCs.

In the past, such a TCRL antibody was selected towards HLA-DR2 molecules complexed with an immunodominant myelin basic protein (MBP) peptide (residues 85–99) and was used for visualization of MBP T-cell epitopes in multiple sclerosis lesions (9). Therapeutic activity of TCRL antibodies in vivo for the prevention or treatment of established EAE was never demonstrated.

Previous work in our laboratory had shown that the anti-HLA-DR4/glutamic acid decarboxylase (GAD)$_{555–567}$ TCRL antibody,

directed towards a peptide derived from type 1 diabetes auto-antigen GAD, was able to block the T-cell response in vivo (10).

Herein, we present the first in vivo proof of concept that a TCRL antibody can cause significant regression of an established autoimmune disorder in a humanized transgenic mouse model of EAE. We demonstrate our working hypothesis that inducing tolerance to myelin antigens would have an effect on EAE initiation and progression and show that the TCRL antibody exerts its biological activity through a combined action of blocking MHC: TCR interactions between the APCs and pathogenic autoreactive CD4$^+$ T cells and modulating specifically CD11b+ APCs that present the autoantigen.

# Results

### Selection and characterization of TCR-like antibodies that recognize myelin oligodendrocyte glycoprotein (MOG)–specific HLA-DR2–restricted autoreactive T-cell epitope

In our experimental setup through the presented studies, we have used in vitro and in vivo systems of human HLA-DR2 APCs and a humanized mouse model of EAE, in which immunization of HLA-DR2 transgenic mice with the MS-associated auto-antigen MOG-derived mouse peptide mMOG$_{35–55}$ results in a chronic EAE (11).

In order to isolate TCRL antibodies specific to HLA-DR2/mMOG$_{35–55}$, we first expressed and purified recombinant HLA-DR2 molecules using constructs in which the intracellular domains of the DR-A1∗0101 and DR-B1∗1501 chains were replaced by leucine-zipper dimerization domains for heterodimer assembly (Fig S1A and B). Purified recombinant DR2 complexes were loaded with mMOG peptide or MBP-derived peptide as a control.

Using the purified recombinant DR2/mMOG$_{35–55}$ complexes, we screened a large human naïve Fab antibody phage display library (12) and isolated in several screening campaigns several TCR-like antibodies with variable affinities termed 2G10, I4Z, and I2A all of which recognized specifically recombinant DR2/mMOG$_{35–55}$ complexes. Fab antibody constructs were transformed into a full chimeric human–mouse IgG antibody construct (IgG2a) by fusing the Fab variable domains to the scaffold of murine IgG2a Fc. The chimeric TCRL Abs were expressed in Expi293 cells, and its TCR-like specificity was further tested on recombinant HLA-DR2 complexes displaying mMOG$_{35–55}$ or control MBP peptide (Fig 1A), as well as native complexes presented on HLA-DR2–positive B-cell line (MGAR) loaded with the mMOG$_{35–55}$ peptide as observed by microscopy (Fig 1B) and flow cytometry (Fig 1C). The TCRL Abs did not bind to recombinant complexes (Fig 1A) or to MGAR B cells that present the control MBP peptide (Fig 1B and C). The proper loading of these peptides to the B-cell APCs was confirmed with a conformational sensitive anti-HLA-DR2 antibody (Fig 1C). Dose-dependent binding of the TCRL Ab exhibited saturable and specific binding with apparent EC50 of high-to-medium affinity (10–100 nM) (Fig 1C) supplemented with SPR data that confirmed these affinities (Fig S1C).

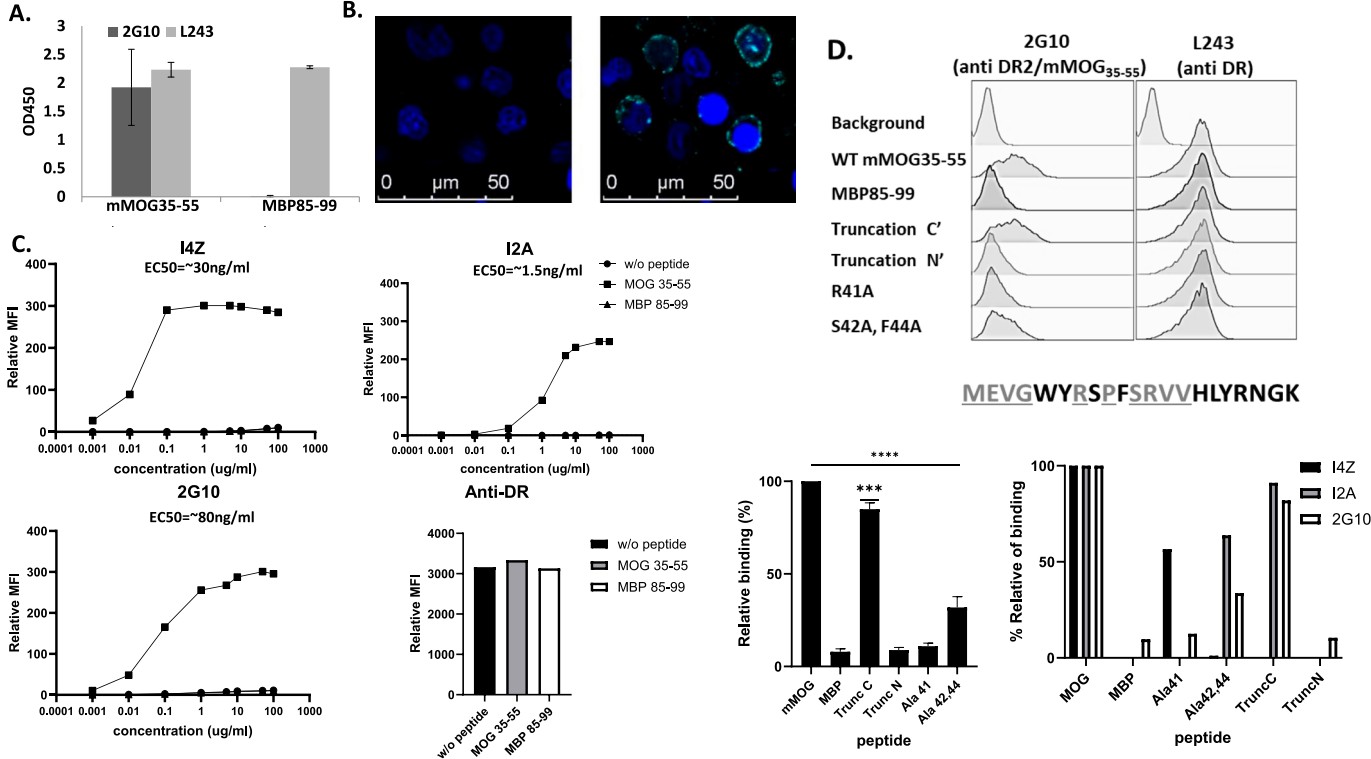

**Figure 1. Isolation and characterization of anti-MOG/DR2 TCRL Abs.**
**(A)** Binding specificity of 2G10 TCRL Ab. ELISA binding assay with 2G10 TCRL Ab towards recombinant HLA-DR2 complexes presenting mMOG$_{35-55}$ or MBP$_{85-99}$ peptides. Anti-DR served as an additional control. **(B)** Immunofluorescence staining of a DR2+ MGAR cell line loaded with mMOG$_{35-55}$ (right) or MBP$_{85-99}$ (left) peptides. Cells were stained with 2G10 TCRL (green). **(C)** MGAR cells were loaded with MOG, MBP, or no peptide. Cells were tested for mIgG2a I4Z, I2A, or 2G10 staining at different antibody concentrations (0.001–100 μg\ml). Anti-DR served as a positive control. **(D)** MGAR cells were loaded with altered MOG$_{35-55}$ peptides, in addition to original MOG$_{35-55}$ and MBP$_{85-99}$. Cells were tested for I4Z, I2A, or 2G10 staining. Top: representative flow cytometry analysis. The amino acid sequence of the wt mMOG$_{35-55}$ peptide is shown. Lower panel: average of three independent experiments examining 2G10 binding (right) or 2G10, I4Z, and I2A (left) normalized to the binding of the WT peptide (100%) and presented as per cent relative binding. Right: statistical analysis was performed by one-way ANOVA and Dunnett's multiple comparison test.

The high specificity of the TCRL Abs, similar to TCR, was validated by their ability to distinguish minor changes in the mMOG$_{35-55}$ peptide sequence. As shown in Fig 1D, we tested the ability of TCRL Abs to recognize DR2-positive APCs loaded with a set of altered mMOG$_{35-55}$ peptide ligands: R41A, S42A+F44A, and N-terminal and C-terminal truncated mMOG (mMOG$_{39-55}$ and mMOG$_{35-48}$, respectively). These mutations did not significantly affect peptide binding to recombinant HLA-DR2 as judged both by competition assays using the biotinylated WT MOG peptide and four altered non-biotinylated peptides (not shown). Truncation at the C-terminus of the peptide did not affect the binding intensity of the I2A and 2G10 Abs compared with their binding to the WT DR2/mMOG$_{35-55}$ but abolished the binding of I4Z. In contrast, the double alanine substitutions at positions 42 and 44 had a significant effect on 2G10 TCRL binding (~70% reduction in binding) and a distinct variable effect on I4Z and I2A. Moreover, alanine substitution at position 41 (R41A) affected binding of 2G10 and I2A, whereas truncation at the N-terminus of the peptide had a detrimental effect and completely (>95%) abolished binding of all three TCRL Abs (Fig 1D). It is important to note that the TCRL Abs do not bind to empty recombinant HLA-DR2 molecules (not shown). These results suggest that TCRL Abs bind the DR2/mMOG$_{35-55}$ epitope in a peptide sequence–dependent manner using multiple specific contact residues that mimic TCR-like fine specificity features.

### TCR-like antibodies can detect EAE disease–specific MOG-derived/HLA-DR2 peptide complexes on the surface of APCs

Next, we evaluated the ability of TCRL Abs to recognize APCs that present HLA-native DR2/MOG complexes, by APC peptide pulsing or naturally occurring antigen processing and presentation. TCRL Abs (shown for 2G10) were able to bind specifically all splenocyte-derived APC subpopulations (B cells, DCs, and macrophages) pulsed with the mMOG$_{35-55}$ peptide (Fig 2A). Upon confirming the specificity of the Abs, we tested whether they can recognize and detect APCs that present mMOG$_{35-55}$ peptide in the context of class II HLA-DR2 molecules after naturally occurring antigen processing and presentation in the context of mMOG$_{35-55}$-induced EAE murine model using HLA-DR2 Tg mice (13). 16 d post-EAE induction, we used TCRL Abs to stain APCs by flow cytometry and detect HLA-DR2/mMOG$_{35-55}$ complexes on the cell surface, investigating which APCs present the autoantigen in spleens and spinal cords harvested from mMOG$_{35-55}$-immunized mice. Although B cells are the most abundant APC population in the spleen, we did not detect presentation of the mMOG$_{35-55}$ peptide on these cells (Fig

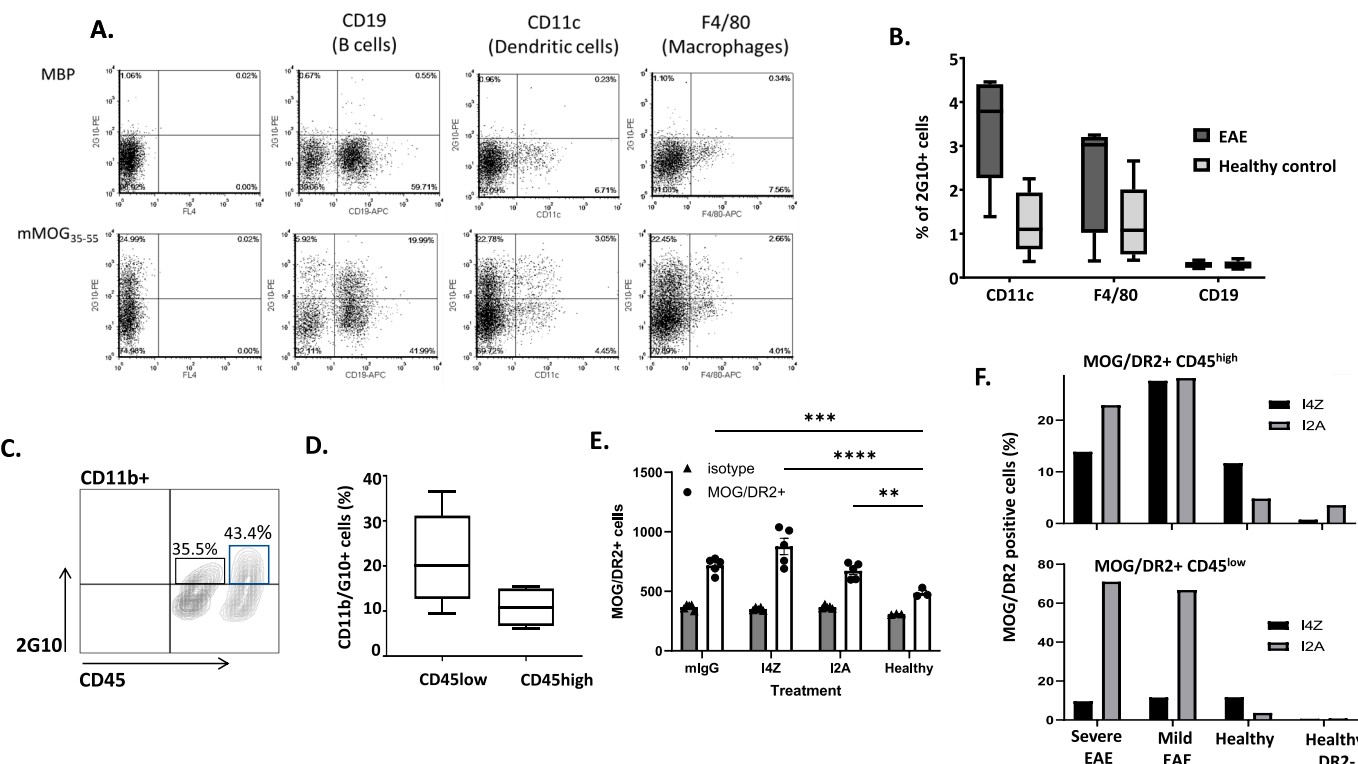

**Figure 2. MOG/DR2 presentation can be detected by anti-MOG/DR2 TCRL Abs within various tissues.**
**(A)** Binding of 2G10 TCRL Ab to various APCs derived from splenocytes of HLA-DR2 Tg mice by flow cytometry. APCs were pulsed with the mMOG$_{35-55}$ and control MBP$_{85-99}$ peptides and subsequently stained with 2G10 TCRL Ab and APC markers CD19, F4/80, and CD11b. **(B)** Average percentage of splenetic DCs (CD11c), macrophages (F4/80), and B cells (CD19) presenting HLA-DR2/mMOG$_{35-55}$ as detected by 2G10 TCRL Ab in EAE-diseased or healthy controls. **(C)** Representative staining of CD11b+ cells in the spinal cord. **(D)** Average percentage of spinal cord CD11b+/CD45$^{high}$ and CD11b+/CD45$^{low}$ that present HLA-DR2/mMOG$_{35-55}$ in EAE mice is shown. **(E)** MOG/DR2 presentation by CD45$^{low}$ monocytes within the CNS of TCRL Ab–treated and control-treated EAE-diseased mice. Healthy DR2 mice served as a control. Presentation was assessed by FACS at experiment endpoint. Statistical analysis was performed by one-way ANOVA and Dunnett's multiple comparison test. **(F)** MOG/DR2 presentation by CD45$^{high}$ (top) or CD45$^{low}$ (bottom) monocytes within the CNS of severely or mildly EAE-diseased mice. Healthy DR2 mice and healthy C57BL/6 DR2(–) mice served as a control. Presentation was assessed by FACS at the experiment endpoint.

2B) but were able to detect significant presentation of mMOG$_{35-55}$ on the CD11c+ DC subpopulation in comparison with a healthy mouse control as expected by their role in antigen uptake (Fig 2B). These TCRL Ab–stained CD11c+ DCs constituted a significant (2–4%) number of the total CD11c+ cells in the spleen of EAE mice compared with controls. Macrophages exhibited some degree of mMOG$_{35-55}$ presentation, which upon statistical analysis was not significant. In contrast to the relatively low frequency of presentation in the periphery, we found substantial presentation of mMOG$_{35-55}$ in the spinal cord of EAE-diseased mice compared with healthy controls, both in the microglial (tissue-resident macrophage) subpopulation (CD11b+/CD45$^{low}$) and in the activated macrophage (CD11b+/CD45$^{high}$) (Fig 2C–F). Although both CD11b+ subpopulations present at high-frequency (10–40%) mMOG$_{35-55}$, we demonstrate that the frequency of presentation of mMOG$_{35-55}$ by CD11b+/CD45$^{low}$ cells is higher compared with CD11b+/CD45$^{high}$ APC subpopulation. Staining data observed for the I4Z and I2A TCRL Abs detecting CD11b+/CD45$^{high/low}$ cells were variable (Fig 2F) in mild and severe EAE but significant compared with healthy mice or HLA-DR2–negative mice (Fig 2E and F).

Investigations with these TCRL Abs enabled the first direct detection of myelin-derived autoantigen presentation in the

CNS and in the periphery during EAE. This approach also identified the APCs involved in autoantigen presentation, as well as the interplay between the high frequency of presentation at the site of inflammation and the leakage of presentation in the periphery.

## The HLA-DR2/MOG-specific TCRL antibodies are functional in vitro and can mediate inhibition of T-cell proliferation and induce antibody- dependent cell cytotoxicity

Next, we examined the biological activity properties of TCRL Abs with the following functions tested: (i) ability to specifically block stimulation of autoreactive T cells as measured by inhibition of proliferation and/or cytokine release and (ii) mediated antibody effector function in the form of ADCC. Inhibition of proliferation in an ex vivo assay is shown for 2G10 (Fig 3A) in which DR2-Tg mice or DR2/MBP-TCR double Tg mice were immunized with mMOG$_{35-55}$ or MBP$_{85-99}$, respectively. 10 d post-immunization, a recall T-cell proliferation assay was performed with immunized mouse–derived CFSE-labelled splenocytes that were restimulated with the priming peptide (mMOG$_{35-55}$ or MBP$_{85-99}$) in the presence of increasing concentrations of the 2G10 TCRL Ab. As shown in Fig 3A, 2G10 TCRL Ab

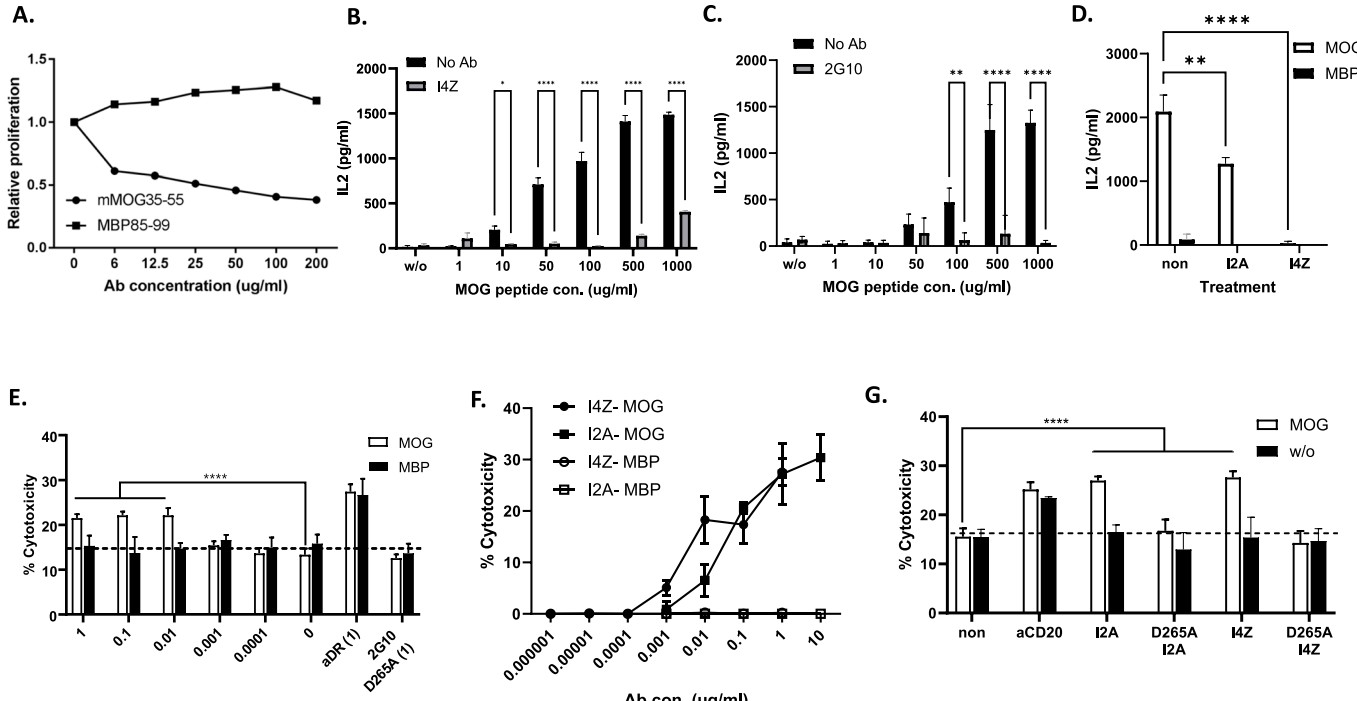

**Figure 3. HLA-DR2/MOG-specific TCRL antibodies are functional in vitro and can mediate inhibition of T-cell proliferation and induce ADCC effector functions.**
**(A)** Antigen-specific inhibition of MOG-specific T-cell proliferation. HLA-DR2-Tg mice were immunized with mMOG$_{35-55}$/CFA, and HLA-DR2/MBP-TCR-Tg mice were immunized with MBP$_{85-99}$/CFA. 10 d post-immunization, CFSE-labelled splenocytes were restimulated with mMOG$_{35-55}$ or MBP$_{85-99}$ peptide, respectively, in the presence of increasing concentrations of 2G10 and 5 d later, CD4 T-cell proliferation was measured. Results were normalized in comparison with proliferation without 2G10 TCRL Ab (relative proliferation). **(B, C, D)** MGAR cells were loaded with various concentrations of the MOG peptide and incubated as described with 10 μg/ml of I4Z (B), 2G10 (C), or I2A (D). After 2 h, MOG-directed CAR T cells were added at a 1:1 E:T ratio for 24 h and IL-2 secretion was assessed. **(E, F, G)** [35]S-methionine radioactive or LDH release assays were used to access cytotoxicity with human PBMCs (F) or murine NK cells (E, G) at 10:1 E:T ratio, by 2G10 (E), I4Z and I2A (F, G) TCRL Abs. D265A ADCC dead mutant TCRL constructs and anti-CD20 Abs were used as controls.

inhibited specifically mMOG$_{35-55}$-reactive CD4 T-cell proliferation in a dose-dependent manner but did not affect the proliferation of MBP$_{85-99}$-reactive T cells, demonstrating ex vivo that 2G10 TCRL Ab can specifically block MHC: TCR interactions and potentially mediate antigen-specific immunomodulation of T-cell activation/proliferation. Inhibition of T-cell activation/cytokine release is also shown in Fig 3B and D for I4Z and I2A and Fig 3C for 2G10, demonstrating peptide titration effect on T-cell stimulation and inhibition by TCRL Abs, as well as specificity (Fig 3D), in which no T-cell stimulation was observed with the MBP peptide. Furthermore, specific ADCC function mediated by TCRL Abs were demonstrated as shown in Fig 3E–G. 2G10 (Fig 3E), I4Z and I2A (Fig 3F and G) exhibited specific ADCC function towards MOG-presenting DR2 cells, which was dose-dependent and comparable to a pan-anti-DR or anti-CD20 antibody. We also used an ADCC dead mutant version of the TCRL Abs (D265A mutation (14)) and demonstrated that this version cannot mediate ADCC (Fig 3E and G).

Altogether, our data demonstrate that the TCRL Abs can mediate antigen-specific modulation and functions towards APCs and T cells exerting the two postulated modes of actions: blocking T-cell functions by competition with TCRs through binding of the TCRL Ab to prevent stimulation/activation of pathogenic T cells; and effector function through antigen-specific ADCC on APCs that present the autoantigen.

## TCRL antibodies prevent EAE in HLA-DR2 humanized murine model

As a first step in examining the potential of TCRL Abs as a therapeutic modulatory agent, we tested their ability to prevent mMOG$_{35-55}$-induced EAE in the DR2 Tg mice. In this EAE prevention experiment, EAE was induced by immunization with mMOG$_{35-55}$ in CFA and pertussis toxin (Ptx). On days 0 and 2 post-immunization, mice were treated with 200 μg 2G10 mIgG, a control mIgG, or vehicle (PBS) (I.P. injection). As shown in Fig 4, TCRL Abs were able to ameliorate EAE severity, with 2G10 TCRL Ab (Fig 4A–C) having the most significant effect in decreasing EAE scores compared with the control groups. I2A was also functional with statistical significance of several independent experiments with overall response (score below 0.5) in 8 out of 13 (62%) treated mice with I2A compared with 3 out of 12 in the control mIgG group that exhibited a disease score of 1 (representative experiment in Fig 4A). 13 d post-immunization, when the control group reached the peak of disease with an average disease score of 2.1 and 2.2 in the mIgG or PBS controls, respectively, most (12/16, 75%) of the 2G10-treated mice had an EAE score ≤ 0.5. 3 of 16 (19%) 2G10-treated mice had an EAE score of 1, and only one of the 2G10-treated mice did not respond to the TCRL Ab treatment and developed a severe disease (EAE score = 3) (Fig 4C).

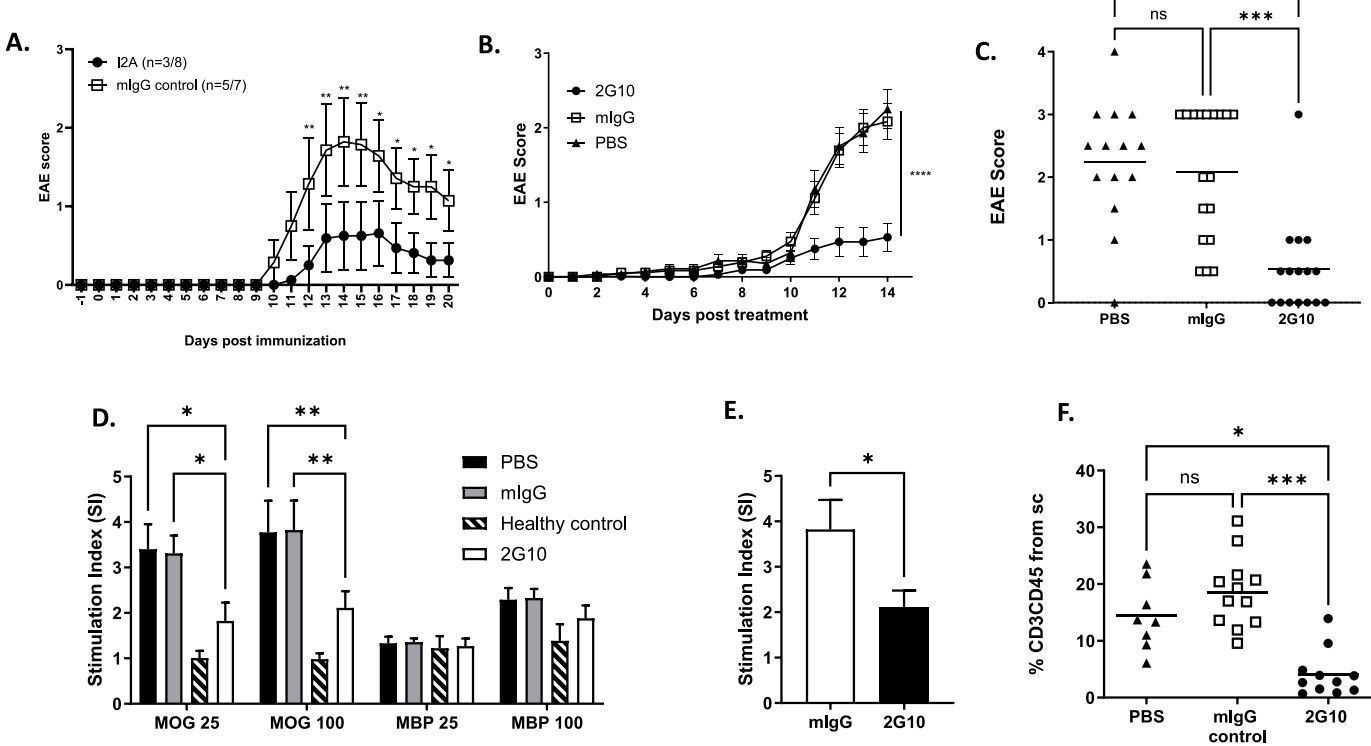

**Figure 4. TCRL Abs prevent EAE in the HLA-DR2 humanized murine model.**
EAE was induced in DR2 male mice by MOG$_{35-55}$/CFA immunization. **(A, B)** Mice were treated i.p. at days -1, 2, 5, and 8 with I2A (A) or at days 0 and 2 with 2G10 (B) TCRL Ab. mIgG or PBS served as a control. **(A, B)** Changes in the EAE score in response to I2A or mIgG (A), or 2G10, PBS, or mIgG (B). **(C)** Individual EAE scores at day 14 post-immunization of (B). **(D, E)** Ex vivo T-cell (D) and CD4 isolated T-cell (E) proliferation in the spleen of EAE-treated mice. Splenocytes from 2G10-treated mice and controls were restimulated with the mMOG$_{35-55}$ priming epitope or a control MBP$_{85-99}$ peptide (25 or 100 μg/ml), and proliferation was monitored by [$^3$H]-thymidine uptake. Data were normalized to proliferation without peptide and are presented as stimulation index (SI). **(F)** CD3+/CD45+ cells in spinal cords collected from a subset of the mice in (B) determined by flow cytometry at the peak of disease (day 14) in response to 2G10, mIgG control, and PBS. In all assays, data were compared using the Kruskal–Wallis test followed by Dunn's multiple comparison test or two-way ANOVA followed by Dunnett's multiple comparison test. Error bars, SEM.

In an attempt to study the mechanism of action of the TCRL Ab and to test whether the prevention of EAE by 2G10 was facilitated by blocking in vivo priming of mMOG$_{35-55}$-specific T cells, we examined recall response to mMOG$_{35-55}$ peptide of splenocytes derived from 2G10-treated mice versus mIgG control or PBS-treated mice. At day 14 post-mMOG$_{35-55}$ peptide immunization, splenocytes were harvested and restimulated with the priming mMOG$_{35-55}$ peptide or the MBP$_{85-99}$ peptide. As shown in Fig 4D, splenocytes from mice treated with 2G10 TCRL Ab exhibited a significantly lower mMOG$_{35-55}$-specific proliferation compared with mIgG control or vehicle-treated mice. Recall proliferation assays with splenocytes primed with MBP$_{85-99}$ peptide did not show significant changes in stimulation between TCRL Ab–treated and control mice (Fig 4D).

The effect on proliferation was governed mostly by CD4 T cells as the inhibitory effect induced by 2G10 TCRL Ab was profound when CD4 was isolated before restimulation with mMOG$_{35-55}$ peptide (Fig 4E). We also examined whether the ability of 2G10 TCRL Ab to inhibit in vivo T-cell priming and proliferation is antigen-specific, by performing recall assays on splenocytes isolated from influenza haemagglutinin-derived (HA) peptide-immunized mice treated with the 2G10 TCRL Ab or vehicle only. In contrast to the significant

blockade of mMOG$_{35-55}$-reactive T-cell priming, we did not observe a difference in HA peptide-mediated proliferation in the 2G10 TCRL Ab–treated versus untreated mice (not shown). These results indicate that in vivo blocking and inhibition of T-cell priming mediated by the 2G10 TCRL Ab are antigen-specific.

Overall, results are in line with our proposed mechanism of action and demonstrate the ability of the 2G10 TCRL Ab to block and inhibit in vivo T-cell priming and proliferation and consequently prevent disease.

Antigen specificity is thus setting the basis of our experimental design to validate the TCRL Ab as a therapeutic modality that can be effective in eliminating or modulating self-response without generally compromising the immune system. To further investigate the biological consequences of 2G10 TCRL Ab treatment in EAE prevention model, we examined migration of T cells and APCs to the CNS. Spinal cord examination of 2G10 TCRL Ab–treated mice revealed a significantly lower level of infiltrating T cells in comparison with mIgG-treated control group, indicating that blocking and inhibition of T-cell priming also inhibited or prevented spinal cord inflammation through reduced T-cell migration into the CNS (Fig 4F). 2G10 TCRL Ab had no influence in EAE prevention model on

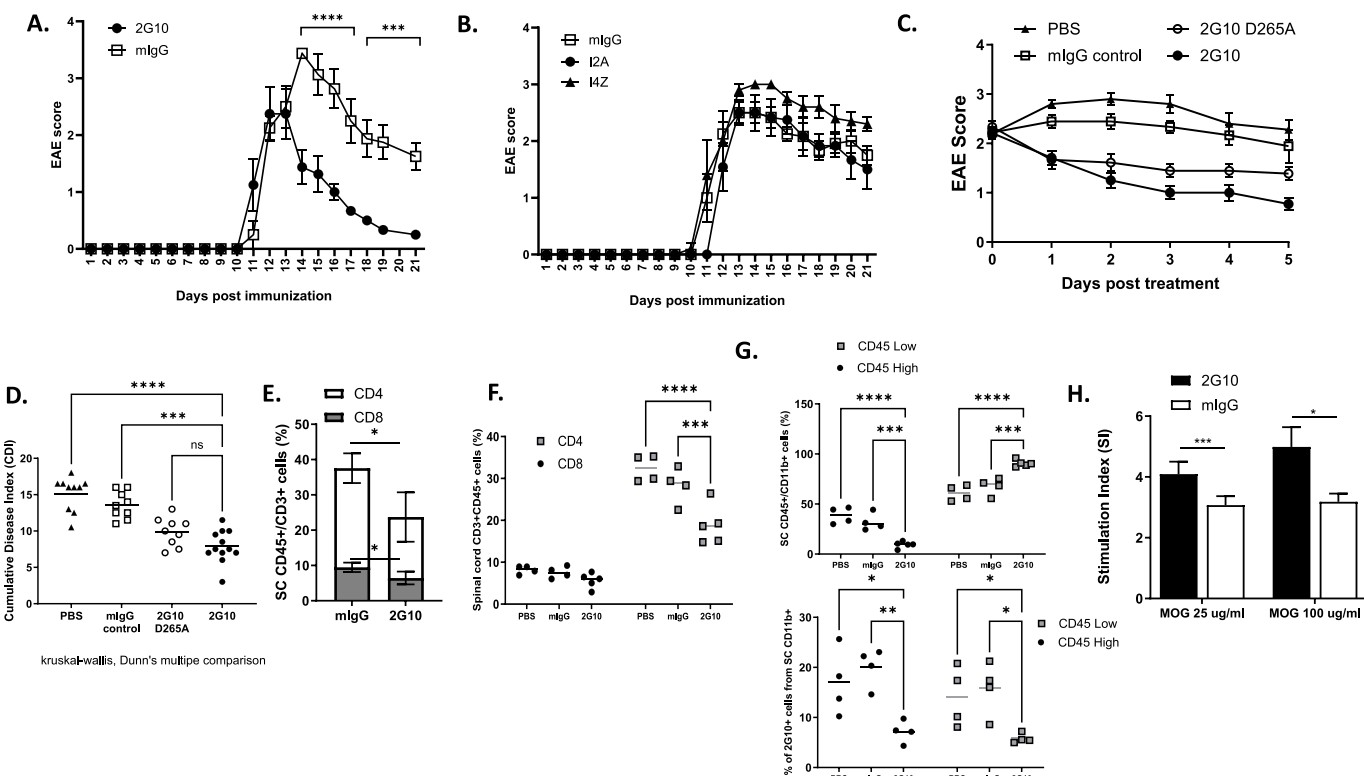

**Figure 5. TCRL Abs modulate established EAE in HLA-DR2 humanized murine model.**
EAE was induced in DR2 male mice by MOG$_{35-55}$/CFA immunization. At a disease score ≥ 2, mice were treated i.p with 200 $\mu$g Ab/mouse, following a second dose after 2 d. **(A, B, C)** Changes in EAE score in response to 2G10 and mIgG (A); I4Z, I2A, or mIgG (B); or 2G10, PBS, mIgG control, and 2G10 D265A (C). **(D)** Individual EAE score of the mice in (C). Data are presented as cumulative disease index (sum of the daily scores since the immunization). **(E, F)** Percentage of CD3$^+$CD45$^+$ cells. **(G)** Top: percentage of mononuclear cells in spinal cords collected from a subset of the mice in (C) determined by flow cytometry 5 d after treatment initiation. Bottom: percentage of CD11b+CD45$^{high}$ and CD11b+CD45$^{low}$ cells presenting the mMOG$_{35-55}$ peptide. Data were compared using the Mann–Whitney $U$ test or the Kruskal–Wallis test followed by Dunn's multiple comparison test or two-way ANOVA followed by Dunnett's multiple comparison test. Error bars, SEM. **(H)** T-cell proliferation in 2G10 TCRL Ab–treated and control EAE mice. Ex vivo proliferative response to restimulation with the mMOG$_{35-55}$ priming peptide determined by [3H]-thymidine uptake in spleens collected from a subset of the mice 5 d after the initiation of treatment. Data were normalized to proliferation without peptide and are presented as stimulation index (SI). In all assays, data were compared using the Mann–Whitney $U$ test or the Kruskal–Wallis test followed by Dunn's multiple comparison test or two-way ANOVA followed by Dunnett's multiple comparison test. Error bars, SEM.

the frequency of CD11b+/CD45$^{high}$ and CD11b+/CD45$^{low}$ APCs in the CNS (not shown). Interestingly, we were able to detect infiltrating CD11b+/CD45$^{high}$ APCs as early as 8 d post-immunization, before the appearance of EAE clinical signs, suggesting that the immunization allowed activated macrophages to cross the blood–brain barrier without additional stimulation/feedback and that 2G10 TCRL Ab was unable to prevent that effect.

### TCRL antibodies modulate established EAE in HLA-DR2 humanized murine model

Antigen-specific TCRL Ab prevention of EAE by blocking T-cell priming, activation and proliferation at the MHC: TCR axis can be useful in the therapeutic window between relapses of MS. However, treatment of an established and ongoing EAE disease has greater implications. To test the ability of TCRL Abs to treat established EAE, we induced chronic EAE in DR2-Tg mice, and began treatment when a clinical score was ≥ 2, usually occurring 10–12 d post-immunization. EAE mice were treated with 2G10 TCRL Ab when a clinical score was ≥2 and 48 h after and were

followed for disease signs. As shown in Fig 5A–D, mice injected with TCRL Abs had variable responses in this model with 2G10 having a significant improvement effect of EAE clinical signs as early as 24 h after the first injection. A representative established EAE treatment experiment over time is shown in Fig 5A, and a summary of two independent established EAE treatment experiments with 2G10 TCRL Ab is shown in Fig 5C and D (day 0 refers to the first dose of 2G10 TCRL Ab when a clinical score is > 2, at 10–12 d post-immunization). Although control mice reached an average clinical score of 3 at the peak of disease, 2G10-treated mice improved gradually, until they reached an average clinical score of 1 (Fig 5A and C). The cumulative disease index (CDI) of individual treated mice was also significantly lower in 2G10-treated mice compared with the CDI of the control mice (Fig 5D). The significantly improved clinical signs in mice treated with the 2G10 TCRL Ab (i.e., a disease score of 1 or even lower) were stable for >50 d post-treatment (not shown). Interestingly, the modulatory effects of I2A and I4Z were not significant and I4Z even enhanced disease scores. These differential effects can be due to multiple factors such as optimal affinity or specific

epitope binding as shown for peptide-dependent binding in TCRL characterization (Fig 1).

A single aspartic acid-to-alanine mutation in the Fc domain has been shown to almost completely abolish the binding of IgG2a and IgG2b to all four classes of Fc gamma receptors and the activation of the complement (14). This mutation was introduced into the 2G10 TCRL Ab, and as shown in Fig 5C and D, the mutant D265A 2G10 TCRL Ab was less efficient than the WT TCRL Ab in treating established EAE, although the differences were not significant. These results may suggest that most of the 2G10 TCRL Ab biological mode of action is related to its ability to inhibit T-cell priming, activation, and proliferation through blocking the MHC:TCR interactions and the pathogenic T-cell:APC axis of interactions. However, the possibility that TCRL Ab activity stems from Fc-mediated ADCC and CDC biological activity cannot be excluded.

Next, we measured and compared infiltration of lymphocytes and active macrophages within the CNS (spinal cord) in EAE-treated versus untreated control mice to determine the influence of 2G10 TCRL Ab on the migration of these cells. As shown in Fig 5E and F, 2G10-treated mice had a significantly lower percentage of T cells in the spinal cord. 2G10 TCRL Ab treatment affected both CD4$^+$ T cells, which were the majority of T cells in the spinal cord, and CD8$^+$ T cells. Not only did the 2G10 TCRL Ab have a significant effect on T-cell frequency in the CNS, but it also exhibited a significant influence on APCs in the CNS. As shown in Figs 5G and S3 for independent experiments, 2G10 TCRL Ab led to a highly significant reduction in the frequency of CD11b+/CD45$^{high}$-activated macrophages in the spinal cord and restored the composition of a healthy spinal cord with a predominant CD11b+/CD45$^{low}$ population of tissue-resident microglial APCs.

Another indication of reduced inflammation in the spinal cord after treatment with the 2G10 TCRL Ab is the reduction in the frequency of APCs that present the autoantigen, that is, the mMOG$_{35-55}$ peptide. As shown in Fig 5G, the reduction in the HLA-DR2/mMOG$_{35-55}$–presenting APCs was observed for both subpopulations of CD11b+/D45$^{high}$ and CD11b+/CD45$^{low}$ cells.

Changes in CD4 and CD8 T-cell infiltrations into spinal cords of 2G10 TCRL Ab–treated mice compared with controls are also shown in Figs S2C and S3B, and the changes in the frequency of CD45$^+$ monocytic APCs and MOG/DR2 presentation by monocytes within the CNS are shown in Figs S2B and D and S3C and D. The measurements of T-cell and APC frequencies were performed in individual mice from large cohorts of mice treated with a disease score ≥ 2 and control mice (Figs S2A and S3A).

Altogether, these results indicate that TCRL Ab–mediated blockade of pathogenic T-cell reactivation is sufficient to suppress an ongoing chronic EAE. Moreover, suppression of T-cell reactivation induced a broad effect on spinal cord inflammation and thus resulted in decreased infiltration of activated APCs into the CNS, as well as a significant reduction in myelin presentation on top of the direct effects on T-cell functions.

In contrast to reduction in the frequency of T cells in the spinal cord after treatment with 2G10 TCRL Ab, we observed an accumulation of mMOG$_{35-55}$-reactive T cells in the spleens of treated mice as demonstrated by the significantly higher mMOG$_{35-55}$-specific proliferation in splenocytes harvested from 2G10 TCRL Ab–treated mice compared with vehicle-treated mice (Fig 5H).

These results suggest that in 2G10 TCRL Ab–treated mice, mMOG$_{35-55}$-specific T cells are sequestered in the spleen and upon the inhibitory effect on their reactivation and proliferative capacity, as well as their ability to migrate to the CNS.

To validate the therapeutic effect of the TCRL antibody in our EAE transgenic model, we conducted a histopathological analysis of DR2 transgenic mice treated with 2G10 TCRL antibody. As shown in Fig 6A, EAE-affected mice treated with control mIgG exhibited significant autoimmune inflammatory demyelination and neurodegeneration in the CNS, as evidenced by lesions indicated by arrows after staining of spinal cord sections for MBP. In contrast, treatment with 2G10 antibody (Fig 6B) resulted in marked histological changes, demonstrating the antibody's ability to induce remyelination and promote recovery from neurodegeneration associated with the disease. These findings offer critical insights into the therapeutic potential of TCRL antibody approaches.

In summary, the results presented herein suggest that antigen-specific targeting of the TCR: MHC axis has potent therapeutic potential and lays grounds for the development of TCRL Ab–based immunotherapeutic agents. These TCRL Abs can be developed for antigen-specific therapy of MS, and could potentially be employed for other autoantigens implicated in various T cell–mediated autoimmune diseases.

Targeting disease-associated APCs that process and present myelin epitopes has the advantage of overcoming the dynamic T-cell repertoire in autoimmune diseases. Moreover, macrophages and myeloid cells at the site of inflammation affect the disease course by releasing oxygen species, and therefore, their targeting may have beneficial effects that are not restricted to their role in T-cell activation. The direct visualization of MOG-presenting APCs at the site of inflammation as shown in this study for EAE disease model makes them a validated target.

## Discussion

The experiments presented in this study demonstrate, for the first time, an in vivo proof of concept for targeting MHC class II–peptide complexes presenting an autoantigen on the surface of APCs that are involved in activation of pathogenic autoreactive T cells in autoimmunity. Such mode of targeting was facilitated by a TCRL Ab that is directed towards the highly characterized MS-associated HLA-DR2–restricted MOG$_{35-55}$ epitope, which was shown to induce severe chronic EAE in HLA-DRA/DRB1*1501-Tg mice (HLA-DR2).

We developed a panel of TCRL antibodies targeting the MOG epitope, distinguished by their unique biochemical properties, particularly their affinity and binding characteristics in terms of kinetics, including on-rates and off-rates (Fig S1). These differences may influence their biological activities. Although all TCRL antibodies demonstrated functional efficacy in vitro—effectively inhibiting T-cell activation and cytokine secretion, as well as mediating ADCC activity—they displayed varying levels of effectiveness in vivo. Notably, in the established EAE disease model, 2G10 antibody exhibited significantly superior activity in ameliorating disease symptoms.

**A. mIgG**

**B. 2G10**

**Figure 6. Histopathological analysis of spinal cord sections from EAE HLA-DR2 Tg mice treated with the 2G10 TCRL Ab.**
**(A, B)** Histopathological analysis of spinal cord sections from EAE HLA-DR2 Tg mice treated with the mIgG control (A) or 2G10 TCRL Ab (B) for evaluation of demyelination (myelin basic protein staining). Mice were treated with the 2G10 TCRL Ab as described in Fig 5A at the experiment endpoint on day 22. Shown are X5, X10, and X40 magnifications. Lesions in control mice (red rows) were not observed in 2G10-treated mice. A staining micrograph represents a single mouse out of 5 with similar staining of 2G10-treated versus mIgG control mice.

We have shown that administration of 2G10 TCRL Ab to MOG$_{35-55}$-induced EAE in HLA-DR2 transgenic mice prevented the disease and most significantly was able to ameliorate established EAE, leading to robust and durable inhibition of disease in animal models.

Elucidating the mode of action of 2G10 revealed that it inhibits the activation of HLA-DR2/mMOG$_{35-55}$-reactive T cells in vivo and reduces the infiltration of autoreactive pathogenic T lymphocytes into the CNS. Moreover, modulation of activated CD11b+ macrophages/microglial APCs that present HLA-DR2/mMOG$_{35-55}$ in the CNS was also demonstrated. Collectively, our data demonstrate the combined action of TCRL Ab in blocking TCR-MHC interactions and modulating APC presentation and activation leading to a profound antigen-specific inhibitory effect on the neuroinflammatory process and regression of EAE.

In previous studies, we have applied our TCRL Ab approach to generate and characterize TCRL Abs against autoreactive T-cell epitopes associated with type 1 diabetes (T1D) and demonstrated that such TCRL Abs are able to block in vivo restimulation of CD4 T cells recognizing T1D-associated autoantigen GAD in the context of class II HLA-DR4 molecules (10, 15). However, the TCRL Ab approach and targeting of autoantigen-associated class II HLA-DR epitopes were never tested in an animal model of autoimmune disease as performed herein for chronic EAE.

TCR-like antibodies was also used to detect specific APCs that present the targeted autoantigen in the spleen and CNS of EAE-diseased mice and thus is a crucial tool for elucidating the mode of action of the TCRL Ab.

The long-term implications of these studies introduce the scientific basis for further evaluation of this approach in preclinical humanized transgenic mouse models of EAE to further demonstrate that the TCRL Ab approach can induce antigen-specific tolerance as a novel immunotherapy for MS and as a proof of concept for other autoimmune and inflammatory diseases.

Class II autoantigen–specific TCRL Abs could confer clinical benefits through three potential modes of actions: (i) blocking TCR-APC interactions at the core disease axis of TCR:MHC; (ii) eliminating autoantigen-presenting APCs, which present not only the targeted epitope or autoantigen-derived MHC class II–peptide complex but also other autoantigens; or (iii) immunomodulating APCs and their microenvironment using the TCRL Ab as a vehicle to deliver an immunosuppressive cytokine (immunocytokine), thus conferring an immunosuppressive environment at the site of inflammation leading to suppression or inhibition of the immune response towards the autoantigen (16, 17, 18, 19, 20, 21, 22).

TCR-like antibodies have shown potential as TCR-mimicking molecules in research and therapeutic development for autoimmune diseases because of their high affinity and ease of engineering. This approach is being developed also in line with studies employing engineered soluble TCRs (23, 24, 25, 26, 27). TCR-like antibodies have been shown to identify specific populations of APCs in autoimmune conditions. For example, in multiple sclerosis (MS) (9), TCRL Abs have identified microglial macrophages as predominant autoreactive APCs in MS lesions. Similarly, in rheumatoid arthritis (RA), TCRL Abs specific for human cartilage

glycoprotein (HC gp-39) epitope presented by HLA-DR4 allele have identified dendritic cells (28) presenting this autoantigen in inflamed joints of RA patients (29). In celiac disease, TCRL Abs have identified plasma cells as the primary cells presenting gluten peptides (30, 31).

Specifically, for MS and to clarify the usefulness of 2G10 TCRL Ab in MS, it should be mentioned about 50% of Caucasian patients are HLA-DR2–positive and thus would limit the utility of this particular TCRL Ab to the allele-specific population and thus would likely require screening of potential recipients for the expression of DR2*1501 before treatment, a process that is common in general in immunotherapy clinical protocols.

The "armed" TCRL Ab approach can target, using single antibody specificity towards a well-defined and characterized autoantigen, APCs that present multiple autoantigens and thus have a profound effect on disease progression. TCRL Ab–induced elimination of autoantigen-specific APCs is expected to suppress the trigger and response against all postulated autoantigens thus changing the course and progression of the disease and may even present a potential solution for antigen spreading.

Despite their potential, TCRL antibodies have not been extensively explored as therapeutics for autoimmune diseases. However, preclinical studies in cancer have demonstrated their therapeutic potential, particularly in targeting intracellular tumour antigens presented by MHC class I molecules (32, 33, 34, 35, 36, 37, 38, 39, 40). A limitation in cancer therapy is the low coverage of TCRL antibodies per cell because of MHC class I down-regulation on tumours. In contrast, MHC class II molecules are typically up-regulated on auto-APCs in autoimmune diseases, providing a more robust target for TCRL antibodies.

In line with the above-mentioned modes of actions, therapeutic strategies for autoimmune diseases using TCRL antibodies could involve depletion of pathology-driving cells, similar to cancer therapy, but with a focus on re-establishing immune balance rather than broad cell elimination. For instance, anti-CD20 antibodies deplete B cells in RA and MS but come with concerns such as long-term benefits and side effects. The use of bispecific antibodies (BsAbs) that could combine the specificity of anti-CD20 and anti-autoantigen peptide/MHC for targeted depletion of pathology-related B cells is of great potential. Non-depleting TCRL mAbs can modulate immune responses by limiting autoantigen–MHC accessibility as shown in this study, thus reducing the activation of cognate T cells. This could be achieved through engineering antibodies with low Fc receptor binding. Autoimmune modulators coupled with TCRL mAbs such as toxins, immunoregulatory cytokines (e.g., IL-10, TGF-$\beta$), or antibodies targeting inflammatory cytokines (e.g., TNF, IL-6, IL-1$\beta$) could be delivered to autoantigen–MHC class II enriched sites to re-establish immune tolerance. TCRL antibodies could be used in chimeric antigen receptor (CAR) formats to construct CAR T cells. In diabetic NOD mice, TCR-like antibodies and CAR T cells expressing an insulin peptide/MHC class II TCRL antibody modulated autoimmunity (41, 42, 43, 44). In another approach, redirecting regulatory T cells (Treg) to the autoimmune milieu could suppress autoreactive effector T cells. Nanobodies with TCR-like specificity can be also developed for various therapeutic modalities (45, 46).

Altogether, the data we present in this work indicate that the biological mode of TCRL Ab action is mostly related to the ability to inhibit T-cell priming, activation and proliferation through blocking MHC:TCR interactions and the pathogenic T-cell:APC axis of inter-actions. This inhibition leads to reduced migration to the inflammatory sites of pathogenic T cells and APCs that present the autoantigen. The possible role of TCRL Ab through Fc-mediated ADCC and CDC biological activity cannot be excluded and needs to be further explored through optimization of TCRL Ab constructs and experimental systems.

Specifically for MS, further investigation of neuroinflammation process as a whole is required to better delineate which inflammatory and neurodegenerative mechanisms are truly distinct but occur in parallel and which are inextricably associated. This analysis will be crucial in aiding the design of more effective therapeutic strategies. A major goal for future treatment of multiple sclerosis may thus be the simultaneous, early targeting of peripheral immune cell function and of CNS-intrinsic inflammation, potentially through combine therapies designed to effectively and specifically modulate these two immunological arms of the disease, along with the provision of neuroprotective or neurodegenerative drugs. The TCRL Ab approach that targets the disease core immune cell functions at the TCR: MHC axis may be a good starting point to develop antigen-specific modulatory agents that will suppress both peripheral immune functions and CNS-associated inflammatory processes.

# Materials and Methods

### Production of recombinant HLA-DR2 in S2 cells

The construction, transfection, and expression of recombinant MHC class II have been described previously 10. Briefly, in these constructs, the intracellular domains of the DR-A1∗0101 and DR-B1∗1501 chains were replaced by leucine-zipper dimerization domains for heterodimer assembly. The Bir-A recognition sequence for biotinylation was introduced to the C-terminus of the DR-A1∗0101 chain. The constructs were cloned between the Bgl II and EcoRI restriction sites into the pMT/BiP/V5-His vector. The vectors were cotransfected with pCoBlast selection vector into S2 cells using FuGENE reagent (Promega). Stable single-cell clones were verified for protein expression. Upon induction of the cells with CuSO4, the supernatant was collected and DR2 complexes were affinity-purified using the anti-DR LB3.1 (ATCC number HB-298) mAb. The purified DR2 complexes were biotinylated by Bir-A ligase (Avidity) and characterized by SDS–PAGE. The biotinylated complexes were loaded with the mMOG$_{35–55}$ peptide or the MBP$_{85–99}$ for 48 h at 37 in PBS, pH 7.4. The right folding of the complexes was verified by recognition of the anti-DR conformation–sensitive mAb (L243) in the ELISA binding assay.

### Selection of phage Abs on biotinylated complexes

Selection of phage Abs on biotinylated complexes was performed as described before. Briefly, a large human Fab library containing

$3.7 \times 10^{10}$ different Fab clones was used for the selection (12). Phages were first preincubated with streptavidin-coated paramagnetic beads (200 μl; Dynabeads) to deplete the streptavidin binders. The remaining phages were subsequently used for panning with decreasing amounts of biotinylated MHC–peptide complexes. The streptavidin-depleted library was incubated in solution with soluble biotinylated DR2/mMOG$_{35-55}$ complexes (20 μg for the first round and 5 μg for the following rounds) for 30 min at room temperature. Streptavidin-coated magnetic beads (200 μl for the first round of selection and 100 μl for the following rounds) were added to the mixture and incubated for 10–15 min at room temperature. The beads were washed extensively 12 times with PBS/0.1% Tween-20, and an additional two washes were with PBS. Bound phages were eluted with triethylamine (100 mM, 5 min at room temperature), followed by neutralization with Tris–HCl (1 M, pH 7.4), and used to infect E. coli TG1 cells (OD = 0.5) for 30 min at 37°C.

## Expression and purification of soluble recombinant Fab Abs

TG1 or BL21 cells were grown to OD600 = 0.8–1.0 and induced to express the recombinant Fab Ab by the addition of isopropylthio-β-galactoside (IPTG) for 3–4 h at 30°C. The periplasmic content was released using the B-PER solution (Pierce), which was applied onto a prewashed TALON column (Clontech). Bound Fabs were eluted using 100 mM imidazole in PBS. The eluted Fabs were dialysed against PBS (overnight, 4°C) to remove residual imidazole.

## Construction of the whole IgG Ab

The H and L Fab genes were cloned for expression as mouse IgG2a Ab into the eukaryotic expression vector pCMV/myc/ER. For the H chain, the multiple cloning site, the myc epitope tag, and the ER retention signal of pCMV/myc/ER were replaced by a cloning site containing recognition sites for BssHI and NheI followed by the mouse IgG2a constant H chain region cDNA isolated by RT–PCR from human lymphocyte total RNA. A similar construct was generated for the L chain. Each shuttle expression vector carries a different antibiotic resistance gene. Expression was facilitated by cotransfection of the two constructs into the Expi293 expression system (Thermo Fisher Scientific). The desired Abs were further purified from their supernatant using protein A affinity chromatography. SDS–PAGE analysis of the purified protein revealed homogenous, pure IgG with the expected molecular mass of ~150 kD.

## ELISA with purified Fab Abs

The binding specificity of individual soluble Fab fragments was determined by ELISA using biotinylated MHC–peptide complexes or peptides. ELISA plates (Falcon) were coated overnight with BSA–biotin (1 μg/well). After being washed, the plates were incubated (1 h at room temperature) with streptavidin (10 μg/ml), washed extensively, and further incubated (1 h at room temperature) with 5 μg/ml of MHC–peptide complexes or peptides. The plates were blocked for 30 min at room temperature with PBS/2% skim milk and were subsequently incubated for 1 h at room temperature with 5 μg/ml soluble purified Fab. After washing, plates were incubated with horseradish peroxidase–conjugated/anti-human Fab antibody. Detection was performed using 3,3′,5,5′-tetramethylbenzidine (TMB; Sigma-Aldrich).

## Spinal cord, brain, and splenocyte isolation

Spleens were removed from euthanized animals under sterile conditions, and single-cell suspensions of leucocytes were prepared by disaggregation of the tissue through a 100-μm nylon mesh (BD Falcon). Cells were washed once with RPMI 1640 supplemented with 10% heat-inactivated FBS (Biological Industries), then incubated with RBC lysis buffer (Sigma-Aldrich) for 5 min to remove red cells, then washed in PBS, and resuspended in culture media.

Brains and spinal cords were passed through 100-μm mesh screens and washed as above. Cells were resuspended in 80% Percoll (GE Healthcare), then overlaid with 40% Percoll to establish a density gradient, and centrifuged at 300 g for 30 min following a method previously described (3). Leucocytes were collected from the resultant interface, counted, and resuspended in FACS buffer (PBS/0.1% BSA), for further analysis.

## Flow cytometry

### For exogenous loading of APCs

A DR2-EBV–transformed B lymphoblast MGAR cell line (IHW#9104) was incubated ON with cRPMI medium containing various concentrations of one of the following peptides, as mentioned: mMOG$_{35-55}$ (MEVGWYRSPFSRVVHLYRNGK), mMOG trunc N (WYRSPFSRVVHLYRNGK), mMOG trunc C (MEVGWYRSPFSRVV), mMOG S42A, F44A (MEVGWYRAPASRVVHLYRNGK), mMOG R41A (MEVGWYASPFSRVVHLYRNGK), or MBP$_{85-99}$ (ENPVVHFFKNIVTPRTP). All peptides were purchased from LifeTein at 95% purity. Cells were washed and incubated with various concentrations of TCRL Abs, or control aDR Ab, as mentioned, for 1 h at 4°C, followed by incubation for 45 min at 4°C with the anti-mouse PE secondary Ab. Cells were then washed and analysed by a FACSCalibur flow cytometer (BD).

### Antibodies

Leucocytes were labelled with a combination of the following antibodies obtained from BioLegend: CD4 APC/Cy7 (GK1.5), CD3 FITC (145-2C11), CD45 APC (30F-11), CD11b PE (M1/70), anti-human HLA-DR PE (L243), anti-mouse IgG PE (Poly4053), Strp-BV, and 7-AAD; eBioscience: CD11c APC (N418) and F4/80 APC (BM8); BD: CD19 APC (1D3); and Jackson ImmunoResearch: anti-human IgG PE.

### Extracellular staining

Single-cell suspensions were washed and resuspended in staining buffer (PBS/0.1% BSA). Fc receptors were blocked with TruStain FcX (anti-mouse CD16/32) (clone 93), and cells were incubated with either hIgG-TCRL Ab or biotin-TCRL, then washed, and incubated with anti-human-BV or Strp-BV, respectively, and conjugated monoclonal antibodies (mAbs) listed above. Unbound mAbs were washed away with staining buffer, and 7-AAD was used to exclude dead cells before flow cytometry analysis using a BD LSR cytometer (BD Biosciences).

## Cytotoxicity assays

Antibody-dependent cell cytotoxicity was measured using a commercially available LDH Non-Radioactive Cytotoxicity Assay (Promega) according to the manufacturer's instructions. Target MGAR cells, loaded with 400 μg/ml MOG or MBP peptide, were incubated for 2 h in the presence of 10 μg/ml mIgG2a TCRL Ab, mIgG2a-D265A TCRL Ab, anti-mouse CD20, and no Ab. After 2 h, all cells were washed and co-incubated with freshly isolated mouse NK cells (using EasySep Mouse NK Cell Isolation Kit [STEMCELL]) at a 10:1 effector-to-target ratio for 4 h, and the concentrations of LDH in supernatants were measured.

For the methionine-based killing assay, target MGAR cells were labelled with $^{35}$S-methionine radioactive isotope for 8–12 h at 37°C and subsequently washed and loaded with 200 μg/ml MOG or MBP peptide overnight. S35-labelled cells were incubated for 2 h in the presence of different concentrations (102-10-6 μg/ml) of hIgG1 TCRL Ab. Anti-human CD20 or no Ab served as a control. After 2 h, all cells were washed and co-incubated with human total PBMCs at a 20:1 effector-to-target ratio for 6 h. The supernatant $^{35}$S-methionine concentration was measured, as an indication for cell death.

Relative cytotoxicity was calculated directly using the following equation.

LDH concentration values were corrected by subtracting culture medium background, effector cell spontaneous release, and target cell spontaneous release. The percentage specific cytotoxicity was calculated as follows:

$$\%Specific\ killing = 100 \times \frac{[Experimental\ value - Effector\ spontaneous\ control - Target\ spontaneous\ control]}{Target\ maximum\ control - Target\ spontaneous\ control}.$$

## IL-2 secretion assay

I4Z anti-MOG/DR2 BW 5143.4 CAR T cells were incubated in the presence of DR2+ MGAR cells loaded with different concentrations of the MOG or MBP peptide, at different E:T ratios, for 24 h. The cell supernatant was collected and tested for IL-2 presence, using the ELISA. Briefly, ELISA plates were incubated in the presence of anti-mouse IL-2 Ab ON, followed by blocking for 30 min at room temperature with PBS/2% skim milk. Subsequently, plates were incubated for 2 h at RT in the presence of a cell culture supernatant. After washing, plates were added with biotinylated anti-mouse IL-2 Ab for 1 h at RT, followed by three washes and incubation with Strep-conjugated horseradish peroxidase for 30 min at RT. Detection was performed using the TMB reagent (Sigma-Aldrich). Commercial mouse IL-2 was used as a positive control and a calibration curve.

## Microscopy

Cells were treated as described above for flow cytometry analysis. Z-sectioning images with a Z-slice of 32.5 μm in 0.5-μm intervals were taken with a laser scanning confocal microscope (LSM 700; Zeiss).

## Mice

DR*1501-Tg mice (a gift from Prof. Vandenbark Portland, Oregon) were bred in-house at the Veterinary Unit, Technion, and used at 8–12 wk of age. All procedures were approved and performed according to institutional guidelines.

## Induction of EAE in DR2-Tg mice

HLA-DR2 mice were screened by FACS for the expression of the HLA transgenes. HLA-DR2–positive male mice between 8 and 12 wk of age were immunized s.c. at two sites on the flanks with 0.2 ml of an emulsion of 100–200 μg immunogenic peptide and complete Freund's adjuvant containing 400 μg of heat-killed *Mycobacterium tuberculosis* H37RA (Difco). In addition, mice were given pertussis toxin (Ptx) from List Biological Laboratories on days 0 and 2 postimmunization (2 × 200 ng per mouse). Immunized mice were assessed daily for clinical signs of EAE on a 5-point scale: 0—clinically normal; 1—decreased tail tone or weak tail only; 2—hindlimb weakness (paraparesis); 3—hindlimb paralysis (paraplegia); 4—weakness of front limbs with paraparesis or paraplegia (quadriparesis); and 5—paralysis of all limbs (quadriplegia). Mean EAE scores and standard deviations for mice grouped according to the initiation of the TCRL Ab or vehicle treatment were calculated for each day and summed for the entire experiment (cumulative disease index, CDI, represents total disease load). Daily mean scores were analysed by two-way ANOVA test for nonparametric comparisons between controls and treated groups.

## Recall proliferation

Splenocytes from immunized mice were isolated as described in reference 8. When indicated, CD4 T cells were isolated using EasySep Mouse CD4 Cell Isolation Kit (STEMCELL). Cells were then cultured in triplicates with or without peptide antigens for 4 d in a stimulation media (DMEM supplemented with 10% FBS, 2 mM sodium pyruvate, 2 mM l-glutamate, 4 mM 2-mercaptoethanol, and 50 μg/ml penicillin/streptomycin), of which the last 16 h was in the presence of 3H-thymidine to assess proliferation responses. For CD4 T-cell cultures, irradiated (3,000 rad) splenocytes from naïve mice were added in a 1:2 ratio. Cultures were harvested on glass fibre filters, and uptake of 3H-thymidine was assessed by liquid scintillation. The stimulation index (SI) was calculated by dividing the cpm of peptide-stimulated cultures by the cpm of control cultures. The statistical significance of the differences between stimulation levels observed in the different treatment groups was determined. Data were compared using the Kruskal–Wallis test followed by Dunn's multiple comparison test.

## CFSE

Isolated splenocytes from immunized mice were stained with CellTrace CFSE Cell Proliferation Kit (Thermo Fisher Scientific) and cultured with or without peptide with different concentrations of 2G10 Ab. After 72 h, CD4$^{+}$ cell proliferation was evaluated by FACS.

## Statistical analysis

We performed a nonparametric Mann–Whitney test or Kruskal–Wallis test followed by Dunn's test for multiple comparisons. EAE score, inhibition, and cytotoxicity assays were analysed using a two-way ANOVA test: ****$P < 0.0001$, ***$P < 0.001$, **$P < 0.01$, and *$P < 0.05$.

# Supplementary Information

# Acknowledgements

The authors are grateful to Profs. Arthur A Vandenbark (Neuroimmunology Research and Multiple Sclerosis Research Laboratory, Oregon Health & Science University, Portland, USA) for the HLA-DR2 transgenic mice and scientific advice and discussions, and Lior Mayo (Tel Aviv University, Israel) for scientific advice and discussions. This work was supported by a grant from the Israel Science Foundation (ISF-1889/20) to Y Reiter.

## Author Contributions

A Goor: conceptualization, data curation, validation, investigation, visualization, methodology, and writing—original draft, review, and editing.
E Altman: conceptualization, data curation, formal analysis, validation, investigation, methodology, and writing—original draft, review, and editing.
I Arman: conceptualization, validation, investigation, visualization, methodology, and writing—original draft, review, and editing.
S Erez: investigation and methodology.
M Haus-Cohen: conceptualization, investigation, methodology, project administration, and writing—original draft, review, and editing.
Y Reiter: conceptualization, resources, supervision, funding acquisition, and writing—original draft, review, and editing.

## Conflict of Interest Statement

The authors declare that they have no conflict of interest.

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
