## [Reviewer comments · Life Science Alliance]

Life Science Alliance

Antigen-Specific modulation of MS/EAE by TCR-like Antibody Targeting Auto-Reactive Epitope

Ilana Goor, Efrat Altman, Inbar Arman, Shir Erez, Maya Haus Cohen, and Yoram Reiter

DOI: <https://doi.org/10.26508/lsa.202402996>

Corresponding author(s): Yoram Reiter, Technion - Israel Institute of Technology

Review Timeline:	Submission Date:	2024-08-14
	Editorial Decision:	2024-08-15
	Revision Received:	2024-10-10
	Editorial Decision:	2024-10-10
	Revision Received:	2024-10-16
	Accepted:	2024-10-23

Transaction Report:

Please note that the manuscript was previously reviewed at another journal and the reports were taken into account in the decision-making process at Life Science Alliance.

Referee #1 Review

Comments on Novelty/Model System for Author:
Please see the comments for the details.

Remarks for Author:

Goor et al. aimed to demonstrate that TCRL antibodies (Abs) can block TCR-MHC interactions, modulate APC presentation and activation, and consequently exert an antigen-specific inhibitory effect on the neuroinflammatory process, leading to regression of Experimental Autoimmune Encephalomyelitis (EAE). The study addresses an important topic with potentially interesting findings. However, the manuscript is not well-written, and the data appear unconvincing and premature. Several major concerns need to be addressed:

1. **Differential Effects of TCRL Abs:** The authors generated three TCRL Abs: 2G10, I4Z, and I2A. Both in vitro and in vivo studies show that these TCRL Abs have different effects. The authors should provide a detailed explanation of why these TCRL Abs exhibit such distinct effects. Are there differences in their binding affinities, specificities, or mechanisms of action?
2. **Clinical Score Concerns:** The clinical score data in Figure 4 indicate that the control mice have a clinical score of around 2, suggesting that EAE is not fully developed. This raises concerns about the adequacy of the disease model used.
3. **Impact on CNS Inflammation:** While Figure 4 suggests that TCRL Abs prevent EAE development in mice, the study lacks data on the impact of these antibodies on CNS inflammation. The authors should present evidence, such as immunohistochemical analysis or flow cytometry data, to show how TCRL Abs affect immune cell infiltration and inflammation in the CNS.
4. **Histopathological Analysis:** EAE is characterized by autoimmune inflammatory demyelination and neurodegeneration in the CNS. The authors should conduct histopathological analyses to determine the effects of TCRL Abs on demyelination and neurodegeneration in the CNS of EAE mice. This would provide crucial insights into the therapeutic potential of these antibodies.
5. **Clinical Relevance to Multiple Sclerosis (MS):** The study's clinical relevance to MS is limited. To enhance this relevance, the authors should investigate whether TCRL Abs can reduce relapse rates in a relapsing-remitting EAE model or attenuate disease progression in a progressive EAE model. Such studies would better mimic the clinical scenarios of relapsing-remitting MS and progressive MS, respectively.

Referee #2 Review

Comments on Novelty/Model System for Author:

This is an outstanding manuscript that presents the first compelling data that a TCR-like (TCRL) antibody specific for the DR2/MOG-35-55 antigen-specific complex can block activation and EAE disease-inducing activity of Class II DR2*1501 restricted MOG-35-55 T cells. As such, the results strongly support the potential of this approach to treat individuals with multiple sclerosis. The experiments are very well controlled and demonstrate that the TCRL antibodies can block TCR-MHC interactions and modulate APC presentation and T cell activation in vitro and in vivo with exquisite antigen-specificity, thus reducing side effects induced by most currently available MS drugs. Of note, the results implicate both CD11c+ dendritic cells and F4/80+ macrophages, but not CD19+ B cells, as relevant Class II MOG-35-55 presenting APC. One suggestion to clarify the usefulness of the 2G10 TCRL in MS would be to discuss the fact that about 50% of Caucasian patients are HLA-DR2 positive and thus would limit the utility of this particular TCRL. Also, it would likely require screening of potential recipients for expression of DR2*1501 prior to treatment.

Remarks for Author:

Regarding Fig. 5, there should be referenced in the text to Fig. 5B in the top paragraph on p15 and Fig. 5F should be referenced in the text in the third paragraph on p17.

Fig. 5I was difficult to follow, since the mean proliferation of the MBP reactive T cells was influenced by two high points that increased the average. Although the difference was found to be significant, the conclusion that these MBP reactive splenocytes responded higher than the unidentified "control" in healthy mice is not compelling, thus weakening the conclusion that the therapeutic effect of the 2G10 TCRL persists in the presence of MBP-reactive T cells. For the sake of clarity, this figure could be deleted, especially since this issue is partially addressed in Figure 3A.

Referee #3 Review

Comments on Novelty/Model System for Author:

The in vitro study is well designed, and carefully executed individually. However, the reviewer does not understand why the authors use this artificial animal model to evaluate the in vivo effect of TCRLs against EAE pathogenesis. Also, the whole manuscript is not necessarily well organized both in the text and the figures, which makes it hard to catch up the results and subsequent arguments. There are numbers of critical issues that makes it difficult to evaluate the manuscript affirmatively and probably need additional experiments to obtain complementary data.

Remarks for Author:

In this work, Goor et al. used a humanized animal model of multiple sclerosis to demonstrate an in vivo proof of concept for targeting class II MHC-peptide complexes to modulate the activation of pathogenic autoreactive T cells. They developed three TCRL Abs with diverse characteristics to investigate their therapeutic potential on humanized EAE model.

The in vitro study is well designed, and carefully executed individually. However, the reviewer does not understand why the authors use this artificial animal model to evaluate the in vivo effect of TCRLs against EAE pathogenesis. Also, the whole manuscript is not necessarily well organized both in the text and the figures, which makes it hard to catch up the results and subsequent arguments. There are numbers of critical issues that makes it difficult to evaluate the manuscript affirmatively and probably need additional experiments to obtain complementary data.

Major comments:

1. The authors claim that their study constitutes an in vivo proof-of-concept for the utility of TCRL as antigen-specific immunomodulators and a therapeutic tool for multiple sclerosis (MS) and other autoimmune and inflammatory diseases. However, the reviewer is suspicious for the scope of this study because of its artificial setting mixing mouse and human immunity. If TCRL used in this study provide an in vivo proof-of-concept, it is applicable for targeting artificial mMOG/HLA-DR2 complex. Although the sequences are known to be highly conserved, MOG peptide (mMOG35-55 and hMOG35-55) is not identical and immune responses against them may differ, as their affinity to HLA-DR2 seems different (Eur. J. Immunol. 2004. 34: 1251-1261). Therefore, it is probably applicable to a very narrow range of the artificial disease model of EAE and is far from in vivo proof-of-concept for general autoimmune diseases. If one wish to prove this in terms of clinical application, one should consider the use of NOG mouse system for example that is reconstituted with human immune cells and analyze EAE against hMOG peptide.

2. The HLA-DR2-transgenic mice used in this study lack the expression of endogenous mouse MHC class II. This must affect much on the physiological development and composition of T cells in adult mice. The reviewer wonders how much does this differential diversity of T cell repertoire affect the

pathogenesis of EAE between the transgenic mice and wt B6 mice. For this reason, the reviewer wonders whether this model is suitable for analyzing whole immune response of EAE pathogenesis, even though this model is useful for estimating specific T cell responses.

3. The authors showed that B cells are a dominant responder to 2G10 TCRL if pulsed with mMOG peptide. However, B cell is a very poor APCs when mice are immunized with mMOG peptide to induce EAE. How are the APCs involved in the pathogenesis and where is the target site of TCRL to inhibit EAE?

4. Although they developed three TCRL Ab with diverse biochemical properties, some of the lack the data of those three TCRL Ab. As a whole, the authors seem to focus on 2G10 TCRL Ab after performing more detailed analysis with it. However, inhibition of proliferation is shown only for 2G10 and more detailed analysis on IL-2 induction is performed on I2A and I4Z TCRL in Fig.3A-C. In addition, the authors just introduce individual data in parallel and do not provide any insight how TCRLs with diverse biochemical properties induce a number of differential in vitro and in vivo effect.

5. MOG is not necessarily a dominant antigen in the context of MS pathogenesis. What is the advantage of MOG/DR2 TCRL against other TCRL including MBL? Also, as self-antigen(s) involved in MS pathogenesis is diverse and not identified yet, what is the future aspect for the TCRL based immune suppression as a therapeutic intervention of autoimmune diseases?

Minor comments:

1. EC50s of each TCRL Ab should be provided in the Fig.1C.
2. Why does the binding of 2G10 to DR2/MOG complex provide such a broad staining in Fig.1D?
3. Replicative EAE data of IA2 treatment are shown as Fig4 A and B, which is obviously redundant. In addition, indication of animal numbers (n=3/8 and so on) is not explained in the legend of Fig4 A/B, which indication is somehow missed in Fig. 4C.
4. In Fig2 F, EAE severity is not clearly defined in either the text or the legend.
5. There is a number of typo and a couple of figures mislabeled in the text that should be corrected.

August 15, 2024

Re: Life Science Alliance manuscript #LSA-2024-02996-T

Prof. Yoram Reiter
Technion-Israel Institute of Technology
Biology
Technion City
Haifa, Israel 3200003
ISRAEL

Dear Dr. Reiter,

Thank you for submitting your manuscript entitled "Antigen-Specific modulation of MS/EAE by TCR-like Antibody Targeting Auto-Reactive Epitope" to Life Science Alliance. We invite you to submit a revised manuscript addressing the Reviewer comments.

Thank you for this interesting contribution to Life Science Alliance. We are looking forward to receiving your revised manuscript.

Sincerely,

B. MANUSCRIPT ORGANIZATION AND FORMATTING:

We have incorporated suggestions made by the reviewers including experimental data and revised the manuscript to the best of our ability addressing the comments.

Referee #1

1. Differential Effects of TCRL Abs: The authors generated three TCRL Abs: 2G10, I4Z, and I2A. Both in vitro and in vivo studies show that these TCRL Abs have different effects. The authors should provide a detailed explanation of why these TCRL Abs exhibit such distinct effects. Are there differences in their binding affinities, specificities, or mechanisms of action?

We have incorporated into the discussion section a paragraph describing the fact that we have selected a panel of TCRL Abs with different biochemical properties. Although all of them exhibited potent activity in in vitro studies they differed in in vivo activity and we provided possible explanations emphasizing the different binding kinetics of the antibodies (page 19).

2. Clinical Score Concerns: The clinical score data in Figure 4 indicate that the control mice have a clinical score of around 2, suggesting that EAE is not fully developed. This raises concerns about the adequacy of the disease model used.

In all experiments performed with the established disease model (Figure 5) we initiated treatment when a fully blown EAE disease was observed, when EAE score was at least 2 and above. In the figure one can observe that the treatment of the specific experiment presented was initiated at disease score of 2.5-3.

This is a score that is defined as established EAE disease as also referenced in the manuscript and the scores are defined in the materials and methods section. In the prevention model (Figure 4) we initiated treatment prior to disease symptoms and disease progressed in controls to a level of 2-3 (see Figure 4B-C). This disease models

has been described before using similar scoring systems and end-points for evaluation (Vandenbark AA, Rich C, Mooney J, Zamora A, Wang C, Huan J, Fugger L, Offner H, Jones R, Burrows GG. Recombinant TCR ligand induces tolerance to myelin oligodendrocyte glycoprotein 35-55 peptide and reverses clinical and histological signs of chronic experimental autoimmune encephalomyelitis in HLA-DR2 transgenic mice. *J Immunol.* 2003 Jul 1;171(1):127-33. doi: 10.4049/jimmunol.171.1.127).

3. Impact on CNS Inflammation: While Figure 4 suggests that TCRL Abs prevent EAE development in mice, the study lacks data on the impact of these antibodies on CNS inflammation. The authors should present evidence, such as immunohistochemical analysis or flow cytometry data, to show how TCRL Abs affect immune cell infiltration and inflammation in the CNS.

We have provided flow cytometry data on the effect of TCRL Abs on immune cells infiltration into the CNS showing the frequency of T cells and APCs infiltration in the spinal cord of TCRL treated versus control mice. In the prevention model (Figure 4F) we show flow staining of CD3+CD45+ cells in the CNS and the established disease model we show (Figure 5E, F) changes in infiltration to CNS of CD4 and CD8 cells as well as changes in CD11b+CD45+ high and low APCs cell populations (Fig. 5G, H), including percent APCs that are stained with the TCRL.

4. Histopathological Analysis: EAE is characterized by autoimmune inflammatory demyelination and neurodegeneration in the CNS. The authors should conduct histopathological analyses to determine the effects of TCRL Abs on demyelination and neurodegeneration in the CNS of EAE mice. This would provide crucial insights into the therapeutic potential of these antibodies.

We performed histopathological analysis of spinal cord samples from 2G10 TCRL treated mice and observed significant differences compared to control. These data are presented in the new Figure 6 and described in the results section (page 18).

5. Clinical Relevance to Multiple Sclerosis (MS): The study's clinical relevance to MS is limited. To enhance this relevance, the authors should investigate whether TCRL Abs can reduce relapse rates in a relapsing-remitting EAE model or attenuate disease progression in a progressive EAE model. Such studies would better mimic the clinical scenarios of relapsing-remitting MS and progressive MS, respectively.

To emphasize the clinical relevance, we specifically chose to generate human antibodies against the human-associated autoantigen. The classical NOD murine model, which was also suggested by Reviewer No. 3, is not applicable in this context because the target antigen in these mice is restricted to the murine class II complex (I-Ag7), which is not relevant to the human system. The HLA-DR2 model, which we used in our study, is the only model that has ever been used for a humanized EAE model in mice. This model has been used in the past specifically for EAE studies, for example, accessing the function of recombinant TCR ligands and using the same MOG 35-55 epitope (see: Recombinant TCR ligand induces tolerance to myelin oligodendrocyte glycoprotein 35-55 peptide and reverses clinical and histological signs of chronic experimental autoimmune encephalomyelitis in HLA-DR2 transgenic mice. *J Immunol.* 2003 Jul 1;171(1):127-33. doi: 10.4049/jimmunol.171.1.127).

Referee #2:

1. One suggestion to clarify the usefulness of the 2G10 TCRL in MS would be to discuss the fact that about 50% of Caucasian patients are HLA-DR2 positive and thus would limit the utility of this particular TCRL. Also, it would likely require screening of potential recipients for expression of DR2*1501 prior to treatment.

We have inserted this valuable comment in the discussion (page 21).

2. Regarding Fig. 5, there should be referenced in the text to Fig. 5B in the top paragraph on p15 and Fig. 5F should be referenced in the text in the third paragraph on p17.

Reference to figures was corrected.

3. Fig. 5I was difficult to follow, since the mean proliferation of the MBP reactive T cells was influenced by two high points that increased the average. Although the difference was found to be significant, the conclusion that these MBP reactive splenocytes responded higher than the unidentified "control" in healthy mice is not compelling, thus weakening the conclusion that the therapeutic effect of the 2G10 TCRL persists in the presence of MBP-reactive T cells. For the sake of clarity, this figure could be deleted, especially since this issue is partially addressed in Figure 3A.

Figure 5I was deleted.

Referee # 3

1. The authors claim that their study constitutes an *in vivo* proof-of-concept for the utility of TCRL as antigen-specific immunomodulators and a therapeutic tool for multiple sclerosis (MS) and other autoimmune and inflammatory diseases. However, the reviewer is suspicious for the scope of this study because of its artificial setting mixing mouse and human immunity. If TCRL used in this study provide an *in vivo* proof-of-concept, it is applicable for targeting artificial mMOG/HLA-DR2 complex. Although the sequences are known to be highly conserved, MOG peptide (mMOG35-55 and hMOG35-55) is not identical and immune responses against them may differ, as their affinity to HLA-DR2 seems different (Eur. J. Immunol. 2004. 34: 1251-1261). Therefore, it is probably applicable to a very narrow range of the artificial disease model of EAE and is far from *in vivo* proof-of-concept for general autoimmune diseases. If one wish to prove this in terms of clinical application, one should consider the use of NOG mouse system for example that is reconstituted with human immune cells and analyze EAE against hMOG peptide.

Similar to our response to reviewer no 1 (comment 5)

To emphasize the clinical relevance, we specifically chose to generate human antibodies against the human-associated autoantigen. The classical NOD murine model, which was also suggested by Reviewer No. 3, is not applicable in this context because the target antigen in these mice is restricted to the murine class II complex (I-Ag7), which is not relevant to the human system. The HLA-DR2 model, which we used in our study, is the only model that has ever been used for a humanized EAE model in mice. This model has been used in the past specifically for EAE studies, for example, accessing the function of recombinant TCR ligands and using the same MOG 35-55 epitope (see: Recombinant TCR ligand induces tolerance to myelin oligodendrocyte glycoprotein 35-55 peptide and reverses clinical and histological signs of chronic experimental autoimmune encephalomyelitis in HLA-DR2 transgenic mice. J Immunol. 2003 Jul 1;171(1):127-33. doi: 10.4049/jimmunol.171.1.127).

2. The HLA-DR2-transgenic mice used in this study lack the expression of endogenous mouse MHC class II. This must affect much on the physiological development and composition of T cells in adult mice. The reviewer wonders how much does this differential diversity of T cell repertoire affect the pathogenesis of EAE between the transgenic mice and wt B6 mice. For this reason, the reviewer wonders whether this model is suitable for analyzing whole immune response of EAE pathogenesis, even though this model is useful for estimating specific T cell responses.

As indicated above we desired for a humanized Tg model rather to a murine one.

Previous studies have demonstrated that this model can be used to evaluate EAE disease close to the human context, for example in testing the impact of EAE with recombinant TCR ligands (see: Recombinant TCR ligand induces tolerance to myelin oligodendrocyte glycoprotein 35-55 peptide and reverses clinical and histological signs of chronic experimental autoimmune encephalomyelitis in HLA-DR2 transgenic mice. J Immunol. 2003 Jul 1;171(1):127-33. doi: 10.4049/jimmunol.171.1.127 and Role of HLA class II genes in susceptibility and resistance to multiple sclerosis: studies using HLA transgenic mice. Luckey D, Bastakoty D, Mangalam AK. J Autoimmun. 2011 Sep;37(2):122-8. doi: 10.1016/j.jaut.2011.05.001.

3. The authors showed that B cells are a dominant responder to 2G10 TCRL if pulsed with mMOG peptide. However, B cell is a very poor APCs when mice are immunized with mMOG peptide to induce EAE. How are the APCs involved in the pathogenesis and where is the target site of TCRL to inhibit EAE?

The studies with B cells as indicated by the reviewer were performed on B cells to demonstrate the TCRL Abs can detect APCs. However, all ex-vivo studies performed with the TCRL Abs on APCs isolated from CNS of diseased mice where on APCs from spinal cords and were shown to be CD11b+/CD45+ glial cells and not B cells. This is described in Figures 4 and 5 as well as the relevant sections referring to these figures in the results section (pages 13-18).

4. Although they developed three TCRL Ab with diverse biochemical properties, some of the lack the data of those three TCRL Ab. As a whole, the authors seem to focus on 2G10 TCRL Ab after performing more detailed analysis with it. However, inhibition of proliferation is shown only for 2G10 and more detailed analysis on IL-2 induction is performed on I2A and I4Z TCRL in Fig.3A-C. In addition, the authors just introduce individual data in parallel and do not provide any insight how TCRLs with diverse biochemical properties induce a number of differential in vitro and in vivo effect.

We performed inhibition assays for IL2 secretion also for 2G10 and they appear in the revised Figure 3C. Added a section in the discussion addressing the TCRL panel properties in terms of diverse biochemical properties provided and possible explanations emphasizing the different binding kinetics of the antibodies (page 19).

5. MOG is not necessarily a dominant antigen in the context of MS pathogenesis. What is the advantage of MOG/DR2 TCRL against other TCRL including MBL? Also, as self-antigen(s) involved in MS

pathogenesis is diverse and not identified yet, what is the future aspect for the TCRL based immune suppression as a therapeutic intervention of autoimmune diseases?

MOG was used before in this EAE HLA-DR2 Transgenic model and can induce efficiently disease in these mice. This was the major reason for selection (see: Recombinant TCR ligand induces tolerance to myelin oligodendrocyte glycoprotein 35-55 peptide and reverses clinical and histological signs of chronic experimental autoimmune encephalomyelitis in HLA-DR2 transgenic mice. J Immunol. 2003 Jul 1;171(1):127-33. doi: 10.4049/jimmunol.171.1.127).

In addition, MOG was reported to be one of the encephalitogenic epitopes T cells recognize and it is an immunodominant epitope among others (epitopes 1-22, 35-55 and 92-106 located at the dimer interface) as foreign antigens and cause the destruction of myelin (demyelination) leading to the clinical condition known as multiple sclerosis (MS) (see Med Chem 2018 14(2):120-128. doi: 10.2174/1573406413666170906123204.Myelin Oligodendrocyte Glycoprotein and Multiple Sclerosis)

6. Minor comments:

- EC50s of each TCRL Ab should be provided in the Fig.1C.

EC50 values were added

- Why does the binding of 2G10 to DR2/MOG complex provide such a broad staining in Fig.1D? **The staining in Fig. 1D was on peptide -loaded cells. There is a large variability in loading efficiency using the loaded APCs which results in broad staining with the TCRL Ab.**

- Replicative EAE data of IA2 treatment are shown as Fig4 A and B, which is obviously redundant. In addition, indication of animal numbers (n=3/8 and so on) is not explained in the legend of Fig4 A/B, which indication is somehow missed in Fig. 4C.

Figure 4B was deleted. Animal numbers are indicated now in the text.

- In Fig2 F, EAE severity is not clearly defined in either the text or the legend.

EAE severity score was detailed in the M@M section related to EAE induction in DR2 Tg mice.

- There is a number of typo and a couple of figures mislabeled in the text that should be corrected.

Manuscript was screened for typo and mislabeled figures

We hope that the changes we introduced into the manuscript address the reviewers comment and that the manuscript will be suitable for publication.

October 10, 2024

RE: Life Science Alliance Manuscript #LSA-2024-02996-TR

Prof. Yoram Reiter
Technion - Israel Institute of Technology
Biology
Technion City
Haifa, Israel 3200003
Israel

Dear Dr. Reiter,

Thank you for submitting your revised manuscript entitled "Antigen-Specific modulation of MS/EAE by TCR-like Antibody Targeting Auto-Reactive Epitope". We would be happy to publish your paper in Life Science Alliance pending final revisions necessary to meet our formatting guidelines.

- please be sure that the authorship listing and order is correct
- please upload your main manuscript text as an editable doc file
- please add ORCID ID for corresponding author-you should have received instructions on how to do so
- please add the author contributions and a conflict of interest statement to the main manuscript text
- please use the [10 author names, et al.] format in your references (i.e. limit the author names to the first 10)
- please add your supplemental figure legends to the main manuscript text
- we encourage you to list the panels in your figure legends in alphabetical order (please correct the figure legend for Figure 2)
- on page 12, you have a callout for Figure E, F, G without a figure number designated; please correct
- please add a figure callout for Figure 2E; Figure 4A; Figure S2; and Figure S3 to your main manuscript text
- in your rebuttal you state that for Figure 5, treatment was initiated once the disease score was greater than 3, but in the text and figure legend you state that it was initiated when the score was greater than or equal to 2. Please clarify and correct.

Figure Check:

- please add scale bars to Figure 1B

A. FINAL FILES:

B. MANUSCRIPT ORGANIZATION AND FORMATTING:

Sincerely,

October 23, 2024

RE: Life Science Alliance Manuscript #LSA-2024-02996-TRR

Prof. Yoram Reiter
Technion - Israel Institute of Technology
Biology
Technion City
Haifa, Israel 3200003
Israel

Dear Dr. Reiter,

Thank you for submitting your Research Article entitled "Antigen-Specific modulation of MS/EAE by TCR-like Antibody Targeting Auto-Reactive Epitope". It is a pleasure to let you know that your manuscript is now accepted for publication in Life Science Alliance. Congratulations on this interesting work.

DISTRIBUTION OF MATERIALS:

Again, congratulations on a very nice paper. I hope you found the review process to be constructive and are pleased with how the manuscript was handled editorially. We look forward to future exciting submissions from your lab.

Sincerely,
